# All-*trans* Retinoic Acids Synergistically and Beneficially Affect In Vitro Glaucomatous Trabecular Meshwork (TM) Models Using 2D and 3D Cell Cultures of Human TM Cells

**DOI:** 10.3390/ijms23179912

**Published:** 2022-08-31

**Authors:** Megumi Watanabe, Tatsuya Sato, Yuri Tsugeno, Megumi Higashide, Masato Furuhashi, Araya Umetsu, Soma Suzuki, Yosuke Ida, Fumihito Hikage, Hiroshi Ohguro

**Affiliations:** 1Departments of Ophthalmology, School of Medicine, Sapporo Medical University, Sapporo 060-8556, Japan; 2Departments of Cardiovascular, Renal and Metabolic Medicine, Sapporo Medical University, Sapporo 060-8556, Japan; 3Departments of Cellular Physiology and Signal Transduction, Sapporo Medical University, Sapporo 060-8556, Japan

**Keywords:** three-dimensional spheroid cultures, human trabecular meshwork (HTM), brimonidine, α2-adrenergic agonist, TGF-β2

## Abstract

We report herein on the effects of all-*trans* retinoic acid (ATRA) on two-dimensional (2D) and three-dimensional (3D) cultures of human trabecular meshwork (HTM) cells that were treated with transforming growth factor β2 (TGF-β2). In the presence of 5 ng/mL TGF-β2, the effects of ATRA on the following were observed: (1) the barrier function of the 2D HTM monolayers, as determined by *trans*-endothelial electrical resistance (TEER) and fluorescein isothiocyanate (FITC) dextran permeability measurements; (2) a Seahorse cellular bio-metabolism analysis; (3) physical properties, including the size and stiffness, of 3D spheroids; (4) the gene expression of extracellular matrix (ECM) molecules, ECM modulators including tissue inhibitor of metalloproteinases (TIMPs), matrix metalloproteinases (MMPs), tight junction (TJ)-related molecules, and endoplasmic reticulum (ER)-stress-related factors. ATRA significantly inhibited the TGF-β2-induced increase in the TEER values and FITC dextran permeability of the 2D monolayers, while an ATRA monotreatment induced similar effects as TGF-β2. A real-time metabolic analysis revealed that ATRA significantly inhibited the TGF-β2-induced shift in metabolic reserve from mitochondrial oxidative phosphorylation to glycolysis in 2D HTM cells, whereas ATRA alone did not induce significant metabolic changes. In contrast, ATRA induced the formation of substantially downsized and softer 3D spheroids in the absence and presence of TGF-β2. The different effects induced by ATRA toward 2D and 3D HTM cells were also supported by the qPCR analysis of several proteins as above. The findings reported here indicate that ATRA may induce synergistic and beneficial effects on TGF-β2-treated 2D- and 3D-cultured HTM cells; those effects varied significantly between the 2D and 3D cultures.

## 1. Introduction

Glaucomatous optic neuropathy (GON) is a characteristic optic nerve head damage condition that is associated with the progressive loss of the visual field upon elevated levels of intraocular pressure (IOP) and is a well-known major risk factor for glaucoma [1,2]. A possible mechanism responsible for elevated IOP levels involves increased resistance to aqueous humor (AH) outflow, which is presumably caused by deposits of excess levels of extracellular matrix (ECM) molecules at the level of trabecular meshwork (TM) during conventional pathways [3,4,5]. Under physiological conditions, the homeostatic regulation of the synthesis and degradation of ECM molecules within the normal phenotype of the TM are required for maintenance of the IOP at normal levels [6,7]. However, such homeostatic regulations are altered in the case of the glaucomatous TM phenotype. That is, reduced cellularity and the accumulation of ECM deposits within those glaucomatous TM increase resistance to AH outflow, thus resulting in elevated levels of IOPs [8,9]. Possible molecular mechanisms responsible for causing such changes in the TM phenotype, among several growth factors and cytokines, include transforming growth factor-β2 (TGF-β2), which is a primary contributor to increased resistance to outflow in TM [10]. In fact, greatly elevated levels of TGF-β2 (1.94–3.46 ng/mL) have been reported within AH specimens obtained from patients with primary open angle glaucoma (POAG), as compared with those in normal human AH (0.41–2.24 ng/mL) [11,12]. Therefore, it is generally thought that such elevated levels of TGF-β2 promote excessive synthesis and deposition of matrix proteins such as collagen IV, fibronectin, laminin, and elastin in the TM, and that these changes contribute subsequent IOP elevation in cases of glaucoma [13,14,15]. Based upon these observations, several in vitro models using TGF-β2-treated TM cells that replicate the glaucomatous TM phenotype have been reported [16].

A derivative of vitamin A, all-*trans* retinoic acid (ATRA), is a potent regulator of the growth and differentiation of various types of cells via RA nuclear receptors [17,18]. Pharmacologically, ATRA exerts anti-inflammatory activity by inhibiting nuclear factor–κB (NF-κB) signaling [19], as well as by exerting antifibrotic effects through attenuating the action of TGF-β [20]. In fact, a derivative of ATRA has been shown to inhibit TGF-β-induced liver fibrosis by suppressing the collagen (COL) 1A2 gene [21]; additionally, an isomer of ATRA, 9-Cis-retinoic acid (9-Cis-RA), was found to attenuate TGF-β-induced fibrotic changes in cultured human mesangial cells [20]. In addition, such ATRA-induced anti-TGF-β effects have been also considered as potential therapeutic candidates for modulating the TGF-β subconjunctival wound healing that is frequently observed post-surgery [22,23]. However, as of the time of writing this paper, little is known in terms of the drug-induced effects of ATRA on glaucomatous TM, in which TGF-β signaling is also involved in pathogenesis. 

Therefore, to elucidate the effects of ATRA toward glaucomatous TM in this study, we measured the following issues using two-dimensional (2D) and three-dimensional (3D) cultures of human trabecular meshwork (HTM) cells [24]: (1) barrier function by *trans*-endothelial electron resistance (TEER) and fluorescein isothiocyanate (FITC) dextran permeability (2D); (2) mitochondrial and glycolytic functions (2D); (3) the physical characteristics, size, and stiffness (3D); (4) the expression of major extracellular matrix (ECM) molecules COL 1, 4, and 6, fibronectin (FN), α smooth muscle actin (αSMA) and its modulators, tissue inhibitors matrix proteinase (TIMP) 1–4, matrix metalloproteinase (MMP) 2, 9, and 14 (2D and 3D), tight junction (TJ)-related molecules, and several endoplasmic reticulum (ER)-stress-related genes. 

## 2. Results

### 2.1. Effects of ATRA on the Barrier Functions and Cellular Metabolic States of 2D-Cultured HTM Monolayers That Replicate a Single-Sheet Layer of the Human TM

In the current study, we initially investigated the drug-induced effects of ATRA on TGF-β2-treated 2D-cultured monolayers, which mimic a single-sheet layer of POAG human TM [24]. In terms of the barrier function of the 2D-cultured HTM cell monolayers, we measured their TEER and FITC dextran permeability, and the results confirmed that TGF-β2 significantly increased the TEER value and relatively decreased the FITC dextran permeability, as has been reported in previous studies [24,25] (Figure 1). An ATRA monotreatment also indicated that 5 ng/mL TGF-β2 also caused similar but less dramatic effects. Unexpectedly and interestingly, the addition of ATRA significantly suppressed the TGF-β2-induced enhancement in TEER values (Figure 1). Since it was suggested that ATRA greatly affected mitochondrial functions [26] as well as glycolysis [27], we next investigated the issue of whether or not ATRA had any effects on TGF-β2-untreated and -treated 2D HTM cells. Interestingly, such discordant effects of ATRA between TGF-β2-untreated and -treated 2D HTM cells were also observed by a real-time cellular metabolic analysis using a Seahorse bioanalyzer (Figure 2). That is, although treatment with ATRA significantly inhibited the TGF-β2-induced shift in metabolic reserve from mitochondrial oxidative phosphorylation to glycolysis, ATRA alone did not affect cellular metabolism in the 2D-cultured HTM cells.

### 2.2. Effects of ATRA on the Physical Properties of the 3D-Cultured HTM Spheroids Replicating Multiple Sheet Layers of the Human TM 

To elucidate the drug-induced effects of ATRA on human TM, which comprises multiple layers, our recently developed an in vitro 3D HTM spheroid model that replicates such a complex human TM structure [24]. Quite interestingly and in contrast to the results related to the 2D HTM monolayers described above, ATRA induced the same effects; that is, ATRA induced the downsizing and softening of the TGF-β2-untreated and -treated 3D HTM spheroids. These were assumed in an in vitro model, replicating the multiple sheet layers of the human TM architecture [24,25] (Figure 3). These functional and physical analyses indicate that ATRA may exert multiple and diverse effects on HTM cells among the several conditions of the TGF-β2-untreated or -treated cells, and the 2D or 3D cultures.

### 2.3. Effects of ATRA on the Expression of the ECM Proteins and Their Modulators, TIMPs, MMPs, TJ-Related Factors, and ER-Stress-Related Factors of the 2D- and 3D-Cultured HTM Cells

To characterize these issues and the expressions of several possible factors including ECM molecules further, regulatory factors including TIMP and MMP and ER-stress-related factors were evaluated by qPCR and/or immunocytochemistry. In terms of the gene expression of ECM proteins (Figure 4), a TGF-β2-stimulated upregulation was detected in COL1 (2D and 3D) and FN (2D), as was observed in our previous studies [24,25,28]; in contrast, an ATRA monotreatment induced a significant downregulation of COL 6 (2D and 3D) and FN (3D) and an upregulation of αSMA (3D); additionally, the addition of ATRA to TGF-β2 caused a significant downregulation of COL1 (2D), COL6 (2D and 3D), and FN (2D and 3D), and the upregulation of COL4 (2D) and αSMA (3D). An ATRA monotreatment also caused identical changes in these ECM proteins in the 3D HTM spheroids. These alterations in the expression of ECM molecules by TGF-β2 and/or ATRA were also confirmed by immunocytochemistry (Appendix A). As shown in Figure 5, the mRNA expression of TIMPs and MMPs resulted in the following: (1) We observed a significant upregulation of TIMP1 (3D) by ATRA; an upregulation of TIMP1 (2D and 3D) and TIMP2 (2D) by TGF-β2 and ATRA, or a downregulation of TIMP2 and TIMP3 (3D) by ATRA; and a downregulation of TIMP2 and TIMP3 (3D) by TGF-β2 and ATRA, respectively. (2) MMP2 and MMP9 (2D) were substantially upregulated or downregulated by TGF-β2 or ATRA, respectively. (3) MMP 2, 9, and 14 were markedly downregulated by ATRA in the presence or absence of TGF-β2. Similar to these differences between 2D and 3D cell cultures, the mRNA expression of claudin11 (CLDN11), a TJ-related molecule, and spliced X-box binding protein (sXBP), an ER-stress-related factor, by TGF-β2 and/or ATRA was also observed (Figure 6). That is, the gene expression of CLDN11 was significantly downregulated by ATRA (2D) and TGF-β2 and/or ATRA (3D), and that of sXBP was upregulated and downregulated by TGF-β2 (2D) and TGF-β2/ATRA (3D), respectively. Therefore, these collective observations reveal that the beneficial effects of ATRA toward the glaucomatous human TM involve a suppression of the TGF-β2-stimulated increases in (1) the TEER values of the 2D monolayers and (2) the stiffness of the 3D spheroids.

## 3. Discussion

RA and its natural and synthetic derivatives, retinoids, are pivotal dietary factors which regulate cellular differentiation and growth by binding nuclear receptors that are comprised of two families of proteins, namely the retinoic acid receptors (RAR) α, β, and γ, and the retinoid X receptors (RXR) α, β, and γ [29,30]. These ligand-activated nuclear receptors stimulate the transcription of target genes by binding to RA-responsive elements within the promoter regions. RAR and RXR proteins bind both all-*trans*- and 9-*cis*-RA, or only 9-cis-RA, which is recognized as the active factor in cell signaling [29]. Such RA-induced nuclear-receptor-mediated regulations are also required for the physiological and pathological events of ocular tissues. The pathophysiological roles of RA include RA suppression of the TGF-β-induced fibrogenic changes of various tissues, including conjunctiva [20,22,23]. In addition to the well-known fact that TGF-β2 is involved in the pathogenesis of glaucomatous TM [16], the bioinformatic prioritization and functional annotation of candidate genes for POAG in genome-wide association studies (GWASs) identified that RA nuclear receptor signaling and TGF-β blood vessel development were the most significant pathways related to ECM metabolism [31]. In addition, RA alone or combined with a glucocorticoid was reported to induce significant upregulation of myocilin expression within immortalized TM cells, suggesting that RA signaling may also be involved in the pathogenesis of the steroid-induced glaucoma (SG)-related TM [32]. These collective observations strongly suggest that RA signaling may have significant roles within the pathogenesis of glaucomatous TM. In fact, RA is identified within AH [33] based upon the fact that an analysis using reverse-phase high-performance liquid chromatography revealed that the RA level in AH obtained from patients with cataracts was 23.3 +/− 2.3 pmol/mL [34]. In the current study, we demonstrated the significant effects of ATRA toward glaucomatous human TM using 2D and 3D HTM cell culture models based upon the following results: (1) ATRA caused the inhibition of a TGF-β2-induced increase in the barrier functions and metabolic reserve from the mitochondrial oxidative phosphorylation to the glycolysis of 2D HTM cells; (2) ATRA substantially downsized and softened 3D HTM spheroids in the absence and presence of TGF-β2. Therefore, we believe that this study is the first demonstration of the effects induced by ATRA on TGF-β2-treated 2D- and 3D-cultured HTM cells using in vitro glaucomatous TM models; note that those effects varied between the 2D and 3D cultures.

As of the time of writing this study, we do know the details of the underlying mechanisms causing such RA-induced effects toward glaucomatous TM; specifically, their function as a biological barrier of the AH outflow remains unclear. However, among the various biological functions of RA [35], it has been shown to regulate the development and maintenance of the blood–brain barrier (BBB) [36] and the blood–retinal barrier (BRB) [37] by promoting the formation of tight junctions. In addition, Nishikiori et al. reported an interesting result in terms of the relationship between RA and BRB. That is, RAR alpha caused significant reductions in vascular leakage in the diabetic retina by antagonizing the loss of tight junction integrity that is induced by diabetes [38]. These observations corresponded well with our current results, which show that ATRA exerts synergistic and beneficial effects in TGF-β2-treated 2D- and 3D-cultured HTM cells; our results were found through in vitro models that replicate the single- and multiple-sheet barrier structures of the HTM [25,26,39,40,41].

However, the current study has some limitations—the following remain unclear: (1) the reason that ATRA-induced effects observed here were different between the cell culture conditions; (2) which signaling is affected by ATRA within the non-glaucomatous and glaucomatous TM. Interestingly, recent observations that the activation of the RA signaling pathway could suppress pseudoexfoliation (PEX)-associated ECM production and the formation of microfibrillar networks via antagonization of Smad-dependent TGF-β1 signaling [42] may provide significant clues to answer these questions. Therefore, to obtain additional information and a better understanding of ATRA-induced effects in non-glaucomatous and glaucomatous HTM—including POAG, SG, and PEX—additional studies using RNA sequencing to identify possible factors will be required among other approaches in our future research.

In conclusion, our current findings demonstrated a novel finding that ATRA may induce synergistic and beneficial effects in TGF-β2-treated 2D- and 3D-cultured HTM cells, providing fundamental evidence of the therapeutic potential of modulating RA-related nuclear receptor actions within glaucoma.

## 4. Materials and Methods

### 4.1. Preparation of 2D- and 3D-Cultured Glaucomatous Human Trabecular Meshwork (HTM) Cells

Two- and three-dimensional cultures of glaucomatous HTMs were prepared from commercially available certified immortalized HTM cells (Applied Biological Materials Inc., Richmond, BC, Canada) [24] and were examined in the absence and presence of a 5 ng/mL solution of TGF-β2, as reported in a previous study [37]. Briefly, the HTM cells that were used after the 20th passage were maintained in 150 mm 2D culture dishes at 37 °C in high-glucose Dulbecco’s modified Eagle medium (HG-DMEM) containing 10% fetal bovine serum (FBS), 1% L-glutamine, 1% antibiotic–antimycotic, until 90% confluence was reached by changing the medium every other day. These 2D-cultured HTM cells were subjected to analyses by TEER measurements, FITC dextran permeability experiments, and a Seahorse bioanalyzer, as described below.

Alternatively, the collected 2D-cultured HTM cells were resuspended in the same culture medium supplemented with 0.25% methylcellulose whose concentration was adjusted at approximately 20,000 cells in 28 μL, and further cultured in hanging-droplet 3D culture plates (# HDP1385, Sigma-Aldrich Co., St. Louis, MO, USA) to produce 3D spheroids during the 6-day protocol. During the course of the 3D spheroid culture, half of the medium was exchanged with fresh medium daily.

To examine the drug-induced effects of ATRA on the TGF-β2-treated 2D and 3D HTM cells, 10 μM ATRA was supplemented during days 1–6. The concentrations of TGF-β2 [35,37,40] and ATRA were confirmed to be the optimum conditions based upon data reported in previous studies [27,41].

### 4.2. Analyses of the Barrier Functions of the 2D-Cultured HTM Cells by TEER and FITC Dextran Permeability

The barrier functions in the 2D-cultured HTM monolayers in the absence and presence of a 5 ng/mL solution of TGF-β2 and/or 10 mM ATRA were evaluated by TEER and FITC dextran permeability measurements, as shown in a previous study [42]. Briefly, the TEER values were measured using a 12 mm-diameter TEER plate (0.4 μm pore size; Corning Transwell, Sigma-Aldrich) in conjunction with an electrical resistance system (KANTO CHEMICAL Co., Inc., Tokyo, Japan). Alternatively, FITC dextran permeability was estimated by measuring the fluorescence intensity of the amount of FITC that permeated through the membrane over a period of 60 min using a multimode plate reader (Enspire; Perkin Elmer, Waltham, MA, USA) at an excitation wavelength of 490 and an emission wavelength of 530 nm.

### 4.3. Analyses of the Real-Time Cellular Metabolism of the 2D-Cultured HTM Cells by a Seahorse Bioanalyzer

As a real-time cellular metabolic function analysis, oxygen consumption rate (OCR) and extracellular acidification rate (ECAR) of the 2D-cultured HTM cells in the absence or presence of TGF-β2 (5 ng/mL) and/or ATRA (10 μM) were measured using a Seahorse XFe96 Bioanalyzer (Agilent Technologies, Santa Clara, CA, U.S.A.), as described in a recent report [43,44,45]. Briefly, 20 × 10^3^ 2D HTM cells under several conditions—(1) nontreated control, (2) treated with TGF-β2, (3) treated with ATRA, and (4) treated with TGF-β2 and ATRA—were grown in 96-well assay plates. After replacing the culture medium with Seahorse XF DMEM assay medium (pH 7.4, Agilent Technologies, #103575-100) supplemented with 5.5 mM glucose, 2.0 mM glutamine, and 1.0 mM sodium pyruvate, the basal OCR and ECAR values were determined using a Seahorse XFe96 Bioanalyzer and the samples were then further analyzed after supplementation with 2.0 μM Oligomycin (Oligo), 5.0 μM carbonyl cyanide-p-trifluoromethoxyphenylhydrazone (FCCP), 1.0 μM rotenone and antimycin A, and 10 mM 2-deoxy-d-glucose (2-DG). The OCR and ECAR values were normalized to the amounts of proteins per well.

### 4.4. Configuration, Physical Properties, Size, and Solidity—Analyses of the 3D HTM Spheroids

The analyses of the physical properties, mean size, and stiffness of the 3D HTM spheroids were performed as described previously [37,46]. Briefly, the mean sizes of the 3D spheroids were directly measured using an inverted microscope (Nikon ECLIPSE TS2; Tokyo, Japan). Alternatively, for the hardness measurements, a single living 3D spheroid was placed on a 3 mm × 3 mm plate and compressed until reaching a deformation of 50% during a period of 20 s using a micro-compressor (MicroSquisher, CellScale, Waterloo, ON, Canada). The force required (μN) was determined, and force/displacement (μN/μm) was calculated.

### 4.5. Immunolabeling of ECM Proteins in the 2D- and 3D-Cultured HTM Cells

The immunolabeling of the ECM proteins in the 2D- and 3D-cultured HTM cells was evaluated as described in a recent report [47,48]. In brief, individual cells were fixed in 4% paraformaldehyde in phosphate-buffered saline (PBS) overnight, blocked in 3% BSA in PBS for 3 h, and then washed twice with PBS for 30 min. They were then sequentially treated overnight (1) with 1:200 dilutions of primary antibodies; an anti-human COL1, COL4, COL6, or FN rabbit antibody at 4 °C, (2) washed 3 times with PBS for 1 h each, (3) 1:1000 dilutions of a secondary antibody; a goat anti-rabbit IgG (488 nm) with phalloidin (594 nm) and 4′,6-diamidino-2-phenylindole (DAPI) for 3 h, and (4) mounting ProLong Gold Antifade Mountant with a cover glass. Immunofluorescent labeling images were obtained by means of a Nikon A1 confocal microscope using a 20× air objective with a resolution of 1024 × 1024 pixels.

### 4.6. Other Analyses

Quantitative PCR using predesigned specific primers (Appendix A) was performed as describe previously [37]. Statistical analyses were processed using Graph Pad Prism 8 (GraphPad Software, San Diego, CA, USA). Statistical significance was determined as a confidence level greater than 95% by a two-tailed Student’s *t*-test or two-way analysis of variance (ANOVA). Then, Tukey’s multiple comparison test was performed, as described previously [37].

## Figures and Tables

**Figure 1 ijms-23-09912-f001:**
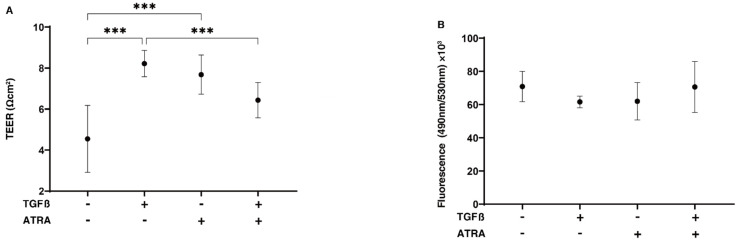
Effects of all-*trans* retinoic acid (ATRA) on trans-endothelial electrical resistance (TEER) values and FITC dextran permeability of TGF-β2-treated 2D cultures of HTM cell monolayers. To evaluate the effects of 10 μM ATRA on the barrier function (Ωcm^2^) of 2D-cultured HTM monolayers, untreated cultures, and cultures that had been treated with 5 ng/mL TGF-β2 (TGFβ), measurements by TEER (**A**) and FITC dextran permeability (**B**) were performed. “+” is reagents addition. “-” is reagents non-addition. All experiments were performed in triplicate using fresh preparations (*n* = 4, total 12). Data are presented as the mean ± standard error of the mean (SEM). *** *p* < 0.005; ANOVA followed by a Tukey’s multiple comparison test.

**Figure 2 ijms-23-09912-f002:**
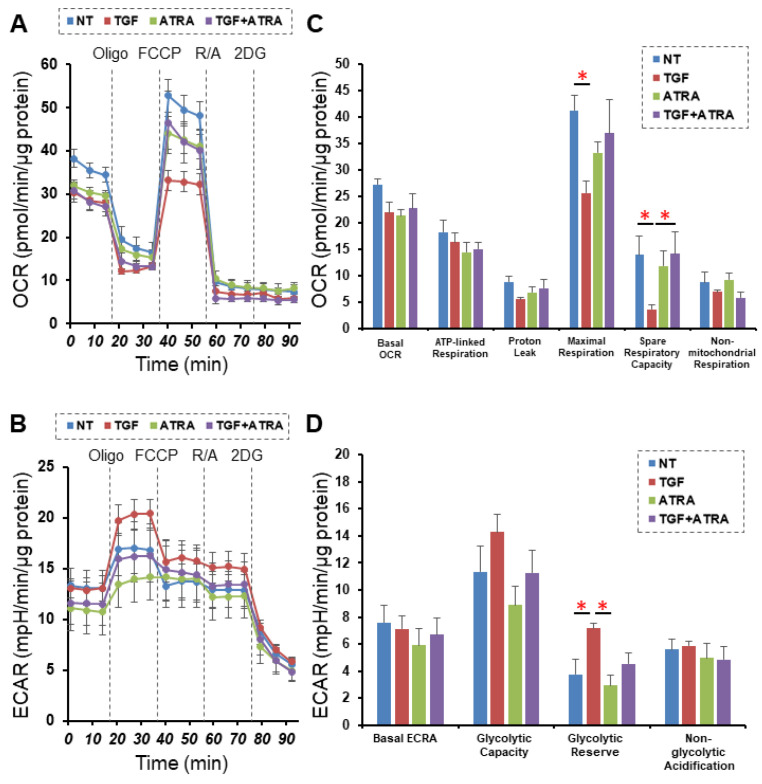
Effects of all-*trans* retinoic acid (ATRA) on a real-time cellular metabolism analysis of 2D-cultured HTM cells. Two-dimensionally cultured HTM cells untreated (NT—nontreated control) or treated with TGF-β2 (TGF, 5 ng/mL) and/or ATRA (10 μM) were subjected to a real-time metabolic function analysis using a Seahorse XFe96 Bioanalyzer. Panel (**A**,**B**): simultaneous measurements of OCR (**A**) and ECAR (**B**) at baseline and those subsequently supplemented with Oligo (a complex V inhibitor), FCCP (a protonophore), rotenone/antimycin A (R/A, complex I/III inhibitors), and 2DG (a hexokinase inhibitor). (**C**,**D**) Key parameters of mitochondrial respiration (**C**) and glycolytic flux (**D**). Basal OCR value was obtained by subtracting OCR with rotenone/antimycin A from baseline OCR. The adenosine triphosphate (ATP)-linked respiration was estimated by subtracting OCR with Oligo from baseline OCR. Proton leak was calculated by subtracting OCR with rotenone/antimycin A from OCR with Oligo. Maximal respiration was calculated by subtracting OCR with rotenone/antimycin A from OCR with FCCP. Spare respiratory capacity was calculated by subtracting baseline OCR from OCR with FCCP. Nonmitochondrial respiration was defined as OCR with rotenone/antimycin A. Basal ECAR was calculated by subtracting ECAR with 2DG from baseline ECAR. Glycolytic capacity was calculated by subtracting ECAR with 2DG from ECAR with Oligo. Glycolytic reserve was calculated by subtracting baseline ECAR from ECAR with oligomycin. Non-glycolytic acidification was defined as the final value of ECAR with 2DG. All experiments were performed in triplicated using fresh preparations (*n* = 5, total 15). Data are presented as the mean ± SEM. * *p* < 0.05; ANOVA followed by a Tukey’s multiple comparison test.

**Figure 3 ijms-23-09912-f003:**
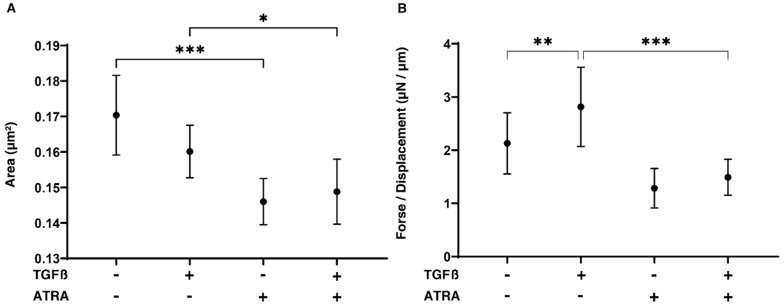
Effects of all-*trans* retinoic acid (ATRA) on the physical properties, size (**A**), and stiffness (**B**) of TGF-β2-treated 3D HTM spheroids. Using nontreated control (NT) cultures and cultures treated with 5 ng/mL TGF-β2 (TGFβ) in the absence or presence of 10 μM ATRA, the mean size areas in the 3D HTM spheroids are plotted in (**A**). Their hardness was estimated by measurement of the force (μN) inducing a 50% semidiameter using a micro-squeezer, and force/displacement (μN/μm) values are plotted in (**B**). “+” is reagents addition. “-” is reagents non-addition. These experiments were performed in triplicate using fresh preparations (*n* = 10, 30 and 15, total 45, for size measurement and stiffness analysis, respectively). Data are presented as the mean ± SEM. * *p* < 0.05, ** *p* < 0.01, *** *p* < 0.005; ANOVA followed by a Tukey’s multiple comparison test.

**Figure 4 ijms-23-09912-f004:**
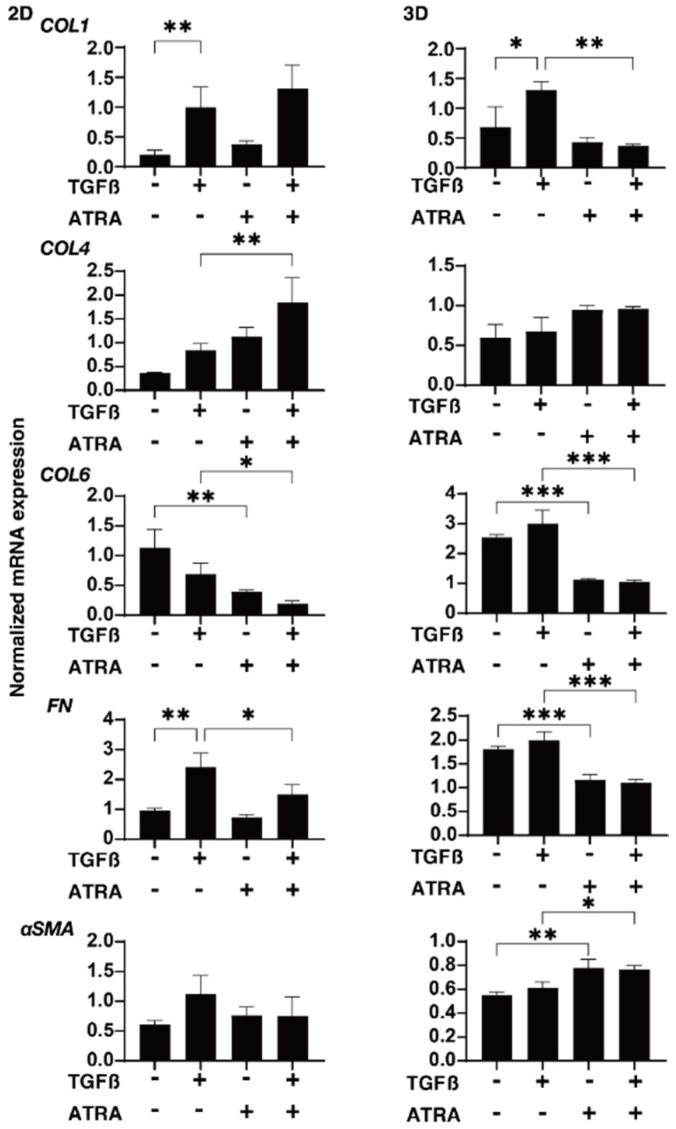
Effects of ATRA on the gene expression of ECM molecules in TGF-β2-treated 2D- and 3D-cultured HTM cells. Two- and three-dimensionally cultured HTM cells untreated (NT) or treated with 5 ng/mL TGF-β2 (TGFβ) and/or ATRA (10 μM) were subjected to quantitative polymerase chain reaction (qPCR) analysis to estimate the expression of mRNA in ECMs (COL1, COL4, COL6, FN, and αSMA). All experiments were performed in triplicate each using freshly prepared 2D HTM cells and 3D HTM spheroids (*n* = 15–20, total 45–60) in each experimental condition. Data are presented as the mean ± SEM. * *p* < 0.05, ** *p* < 0.01, *** *p* < 0.005; ANOVA followed by a Tukey’s multiple comparison test.

**Figure 5 ijms-23-09912-f005:**
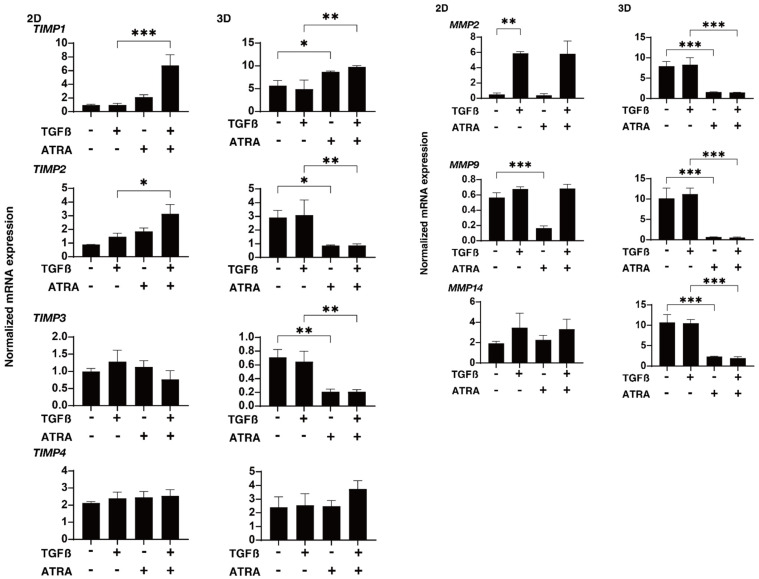
Effects of ATRA on the mRNA expression of TIMPs and MMPs in TGF-β2-treated 2D- and 3D-cultured HTM cells. Two- and three-dimensionally cultured HTM cells untreated (NT; nontreated control) or treated with 5 ng/mL TGF-β2 (TGFβ) and/or ATRA (10 μM) in the absence or presence of 10 μM ATRA were subjected to qPCR analysis to estimate the expression of mRNA in *TIMP1–4*, and *MMP 2, 9,* and *14*. All experiments were performed in triplicate each using freshly prepared 2D HTM cells and 3D HTM spheroids (*n* = 15–20, total 45–60) in each experimental condition. Data are presented as the mean ± SEM. * *p* < 0.05, ** *p* < 0.01, *** *p* < 0.005; ANOVA followed by a Tukey’s multiple comparison test.

**Figure 6 ijms-23-09912-f006:**
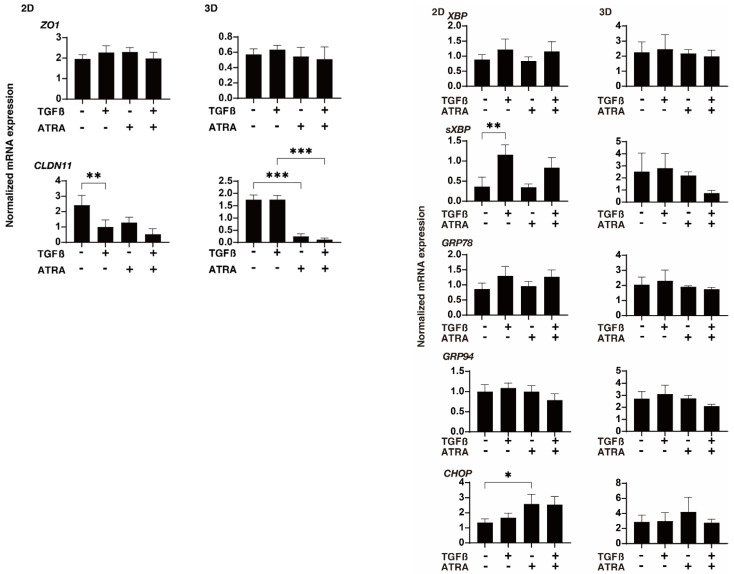
Effects of ATRA on the mRNA expression of TJ-related molecules and major ER-stress-related genes in TGF-β2-treated 2D- and 3D-cultured HTM cells. Two- and three-dimensionally cultured HTM cells untreated (NT) or treated with 5 ng/mL TGF-β2 (TGFβ) and/or ATRA (10 μM) were subjected to qPCR analysis to estimate the expression of mRNA in TJ-related molecules, ZO1 and CLDN11, and major ER-stress-related genes including the X-box binding protein (XBP), sXBP, glucose regulator protein (GRP)78, GRP94, and CCAAT/enhancer-binding protein homologous protein (CHOP). All experiments were performed in triplicate each using freshly prepared 2D HTM cells and 3D HTM spheroids (*n* = 15–20, total 45–60) in each experimental condition. Data are presented as the mean ± SEM. * *p* < 0.05, ** *p* < 0.01, *** *p* < 0.005; ANOVA followed by a Tukey’s multiple comparison test.

## Data Availability

The data that support the findings of this study are available from the corresponding author upon reasonable request.

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
