# Peer review of "All-trans Retinoic Acids Synergistically and Beneficially Affect In Vitro Glaucomatous Trabecular Meshwork (TM) Models Using 2D and 3D Cell Cultures of Human TM Cells"

_ijms, 2022, doi:10.3390/ijms23179912_

Round 1
Reviewer 1 Report
This is an interesting basic science study on in Vito trabecular mesh work cells. The results are encouraging.
Author Response
Dear Editor,
Thank you very much for the constructive comments concerning our manuscript, " All trans-retinoic acids synergistically and beneficially affect the TGF-β2 induced effects on 2 D and 3D cultured human trabecular meshwork (HTM) cells”. We carefully examined all of the Editor and Reviewer comments and prepared a revised version of our paper that takes these comments into account. The changes are listed below.
Reviewer 1
This is an interesting basic science study on in Vito trabecular mesh work cells. The results are encouraging.
Answer; Thank you for such an encouraging and constructive comment.
Reviewer 2 Report
Watanabe et al. investigated the effect of ATRA on 2D and 3D cultures of human trabecular meshwork (HTM) cells treated with TGF-b2. The authors examined the barrier function of the 2D HTM monolayers and the physical properties of the 3D spheroids in cell culture. The authors report that ATRA may synergistically and beneficially affect TGF-b2-treated HTM cells. The finding is significant for the treatment of patients with cataracts. However, the manuscript needs more clarifications before being published in IJJMS—specifically, I could not understand why the effects of ATRA treatment are synergistic and beneficial. The detailed comments are listed below.
- In the results section, please consider including subsections to improve the clarity of studies. In addition, please consider including a schematic to summarize the overall results and interpretations so that readers can easily understand the significance.
- Lines 105-107, please explain more details about the rationale of experimental designs, principles, and anticipated outcomes so that readers can easily understand the logical flow.
- Figure 1 legend, all experiments were performed in triplicate using fresh preparations (n=4). Does this indicate n=12 in total? Same questions to other figure legends.
- Line 165, Supplemental Fig. 1, since it is pseudo colors, please clarify which color indicates which molecule, such as blue indicates DAPI, etc.
- Lines 257-258, please briefly describe a possible explanation of these observed differences.
- Lines 262-263, correctly “siRNA”, and please clarify the sentence. What kinds of genes would future studies include using siRNA? What does “other approaches will be required” indicate? Please consider specifying the information to improve the clarity.
- Lines 280, 285, 289, and 297, Please briefly describe the experimental details on instruments, software, settings, etc.
- Line 299, correctly “4.5 Other analyses.”
- Please consider including a list of abbreviations.
Author Response
Dear Editor,
Thank you very much for the constructive comments concerning our manuscript, " All trans-retinoic acids synergistically and beneficially affect the TGF-β2 induced effects on 2 D and 3D cultured human trabecular meshwork (HTM) cells”. The changes are listed below.
Watanabe et al. investigated the effect of ATRA on 2D and 3D cultures of human trabecular meshwork (HTM) cells treated with TGF-b2. The authors examined the barrier function of the 2D HTM monolayers and the physical properties of the 3D spheroids in cell culture. The authors report that ATRA may synergistically and beneficially affect TGF-b2-treated HTM cells. The finding is significant for the treatment of patients with cataracts. However, the manuscript needs more clarifications before being published in IJJMS—specifically, I could not understand why the effects of ATRA treatment are synergistic and beneficial. The detailed comments are listed below.
- In the results section, please consider including subsections to improve the clarity of studies. In addition, please consider including a schematic to summarize the overall results and interpretations so that readers can easily understand the significance.
Answer; Thank you for this comment. As suggested, the results were divided into 3 subsections; 1) Effects of ATRA on the barrier functions and cellular metabolic states of 2D cultured HTM monolayers that replicate a single sheet layer of the human TM, 2) Effects of ATRA on the physical properties of the 3D cultured HTM spheroids replicating multiple sheet layers of the human TM, and 3) Effects of ATRA on the expression of the ECM proteins and their modulators, TIMPs, MMPs, TJ related factors, and ER-stress related factors of the 2D and 3D cultured HTM cells. In terms of the proposal to include a schematic to summarize the overall results and interpretations, we conclude that this is an excellent idea. Nevertheless, as of this writing, there are several unidentified issues that need to be explored as described in the last sentences of the discussion; “In the present study, to elucidate these unidentified issues, we initiated several studies as above and found that ATRA exerts synergistic and beneficial effects in TGF-b2 treated 2D and 3D cultured HTM cells, which are in vitro models that replicate single and multiple sheet structures of the HTM [25,27,37,40,48]. The findings indicated that the ATRA induced effects observed here were different between the cell culture conditions, 2D and 3D as well as the glaucomatous and none-glaucomatous HTM phenotypes. However, as of this writing, we are unable to confirm the mechanisms responsible for causing such cell culture dependent diversity in terms of the ATRA induced effects. Therefore, to obtain additional information and a better understanding of ATRA-induced effects in glaucomatous and none-glaucomatous HTM, additional studies using RNA sequencing to identify possible factors, and other approaches will be required as our next future project.”. Therefore, in this situation, I hesitate to include ambiguous schema that may cause misunderstanding as well as overstatement.
- Lines 105-107, please explain more details about the rationale of experimental designs, principles, and anticipated outcomes so that readers can easily understand the logical flow.
Answer; Thank you for this comment. As suggested, a rationale of the study purpose in terms of the Seahorse analyzed was included; “Since it was suggested that ATRA greatly affected mitochondrial functions [38] as well as glycolysis [39], we next investigated the issue of whether or not ATRA had any effects on TGF-b2 untreated and treated 2D HTM cells. Interestingly, such discordance effects by ATRA between TGF-b2 untreated and treated 2D HTM cells were also observed by a real-time cellular metabolic analysis using a Seahorse Bioanalyzer (Fig. 2).”.
- Figure 1 legend, all experiments were performed in triplicate using fresh preparations (n=4). Does this indicate n=12 in total? Same questions to other figure legends.
Answer; Thank you for this comment. Yes, it does. This information is included in the figure legends.
- Line 165, Supplemental Fig. 1, since it is pseudo colors, please clarify which color indicates which molecule, such as blue indicates DAPI, etc.
Answer; Thank you for this comment. As suggested, each color was clarified within the legend; “2D and 3D HTM cells untreated and treated with TGF-β2 (TGFβ, 5 ng/ml were subjected to immunolabeling for COL 1, COL 4, COL 6, FN and aSMA in the presence or absence of 10 mM ATRA. Experiments were repeated in duplicate (n=5 in each, total n=10). Representative merged images of ECM staining (green) with DAPI (blue) and Phalloidin (red) are shown in panels A (2D) and B (3D).”.
- Lines 257-258, please briefly describe a possible explanation of these observed differences.
- Lines 262-263, correctly “siRNA”, and please clarify the sentence. What kinds of genes would future studies include using siRNA? What does “other approaches will be required” indicate? Please consider specifying the information to improve the clarity.
Answers for #5 and #6; Thank you for these comments. As suggested, the last sentences of the discussion were complicated and could lead misunderstandings. Therefore, those sections were rewritten in a more simpler manner; “In the present study, to elucidate these unidentified issues, we initiated several studies as above and found that ATRA exerts synergistic and beneficial effects in TGF-b2 treated 2D and 3D cultured HTM cells, which are in vitro models that replicate single and multiple sheet structures of the HTM [25,27,37,40,48]. The findings indicated that the ATRA induced effects observed here were different between the cell culture conditions, 2D and 3D as well as the glaucomatous and none-glaucomatous HTM phenotypes. However, as of this writing, we are unable to confirm the mechanisms responsible for causing such cell culture dependent diversity in terms of the ATRA induced effects. Therefore, to obtain additional information and a better understanding of ATRA-induced effects in glaucomatous and none-glaucomatous HTM, additional studies using RNA sequencing to identify possible factors, and other approaches will be required as our next future project.”
- Lines 280, 285, 289, and 297, Please briefly describe the experimental details on instruments, software, settings, etc.
Answer; Thank you for this comment, As suggested, additional details concerning the corresponding parts of the methods are now included; “Analyses of the barrier functions of the 2D cultured HTM cells by TEER and FITC-dextran permeability. The barrier functions in the 2D cultured HTM monolayers in the absence and presence of a 5 ng/mL solution of TGF-β2 and/or 10 mM ATRA were evaluated by TEER and FITC dextran permeability measurements as shown in a previous study [30]. Briefly, the TEER values were measured using a 12 mm diameter TEER plate (0.4 μm pore size; Corning Transwell, Sigma-Aldrich) in conjunction with an electrical resistance system (KANTO CHEMICAL CO. INC., Tokyo, Japan). Alternatively, FITC-dextran permeability was estimated by measuring the fluorescence intensity of the amount of FITC that permeated through the membrane over a period of 60 min using a multimode plate reader (Enspire; Perkin Elmer, MA USA) at an excitation wavelength of 490 and an emission wavelength of 530 nm.”, “Analyses of the real time cellular metabolism of the 2D cultured HTM cells by a Seahorse Bioanalyzer. As a real-time cellular metabolic function analysis, OCR and ECAR of the 2D cultured HTM cells in the absence or presence of TGF-β2 (5 ng/mL) and/or ATRA (10 mM) were measured using a Seahorse XFe96 Bioanalyzer (Agilent Technologies, Santa Clara, CA, U.S.A.) as described in a recent report [31-33]. Briefly, 20 x 103 2D HTM cells under several conditions; 1) non-treated control, 2) treated with TGF-β2, 3) treated with ATRA and 4) treated with TGF-β2 and ATRA were grown in wells of a 96-well assay plate. After replacing the culture medium with Seahorse XF DMEM assay medium (pH 7.4, Agilent Technologies, #103575-100) supplemented with 5.5 mM glucose, 2.0 mM glutamine, and 1.0 mM sodium pyruvate, the basal OCR and ECAR values were determined using a Seahorse XFe96 Bioanalyzer and the samples were then further analyzed after supplementation with 2.0 μM oligomycin, 5.0 μM FCCP, 1.0 μM rotenone and antimycin A, and 10mM 2-DG. The OCR and ECAR values were normalized to the amount of protein per well.”, “Configuration, physical properties, size and solidity, analyses of the 3D HTM spheroids. The analyses of the physical properties, mean size and stiffness of the 3D HTM spheroids were performed as described previously [25,34]. Briefly, the mean sizes of the 3D spheroids were directly measured using an inverted microscope (Nikon ECLIPSE TS2; Tokyo, Japan). Alternatively, for the hardness measurements, a single living 3D spheroid was placed on a 3-mm × 3-mm plate and compressed until reaching a deformation of 50 % during a period of 20 seconds using a micro-compressor (MicroSquisher, CellScale, Waterloo, ON, Canada). The force required (μN) was determined, and force/displacement (μN/μm) was calculated.”, and “Immunolabeling of ECM proteins in the 2D and 3D cultured HTM cells. The immunolabeling of the ECM proteins in the 2D and 3D cultured HTM cells was evaluated as described in a recent report [35,36]. In brief, individual cells were fixed in 4 % paraformaldehyde in PBS overnight, blocked in 3 % BSA in PBS for 3 hours and then washed twice with PBS for 30 minutes. They were then sequentially treated overnight with 1) 1:200 dilutions of primary antibodies; an anti-human COL1, COL4, COL6 or FN rabbit antibody at 4 °C, 2) washed 3 times with PBS for 1 hour each, 3) 1:1000 dilutions of a secondary antibody; a goat anti-rabbit IgG (488 nm) with phalloidin (594 nm) and DAPI for 3 hrs, and 4) mounting ProLong Gold Antifade Mountant with a cover glass. Immunofluorescent labeling images were obtained by means of a Nikon A1 confocal microscope using a ×20 air objective with a resolution of 1024 × 1024 pixels.”.
- Line 299, correctly “4.5 Other analyses.”
Answer; Thank you for this comment. As pointed out, this was changed to “4.5 Other analyses.”
- Please consider including a list of abbreviations.
Answer; Thank you for this comment. As suggested, a list of abbreviations was prepared as Table 1.
Reviewer 3 Report
Title: All Trans-Retinoic Acids Synergistically and Beneficially Af- 2 fect the TGF-β2 Induced Effects on 2 D and 3D Cultured Hu- 3 man Trabecular Meshwork (HTM) Cells
It is interesting experimental study on TM cells.
Major comments
1. Method is not descriptive, Method 4.2 to be more detail lines 285, Method 4.3 to be more detail. Line 289.
2. High Plagiarism rate: issue 19% it should be minimized.
3. In Discussion: There are numerous conclusions that cannot be drawn from the outcomes of this study. It should contain the analysis of metabolism data and ATRA _TGFB effects, SMA, etc. Association of eye development and vitamin A and immunity of Treg are gap with this study results.
Minor comments:
1. English editing service required
For example
lines 42-43: IOP is well known risk factor for GON, not GON is well known- risk factors
lines 49-52: “However, and in contrast, such homeostatic regulations are altered in the case of the glaucomatous TM phenotype, in which reduced cellularity and the accumulation of ECM deposits are observed, and the resistance to AH outflow is increased thus resulting in the elevation of IOPs.”
lines 73-75: “However, and in contrast, as of this writing, our knowledge of the drug induced effects of ATRA toward glaucomatous TM, in which TGF-β signaling is also involved in the pathogenesis, remains quite limited.”
The sentence is complex, required scientific expression to be understand.
Abbreviation repetition, line 64 all-trans-retinoic acid (ATRA), line 88 all-trans-retinoic acid (ATRA)
Lines 77, 91 trans-endothelial electron resistance (TEER)
2. Correct expression and need references
Lines 62-73: Based upon these concepts, using TGF-β2 treated TM cells, several in vitro models that replicate the glaucomatous TM phenotype have been reported.
Author Response
Dear Editor,
Thank you very much for the constructive comments concerning our manuscript, " All trans-retinoic acids synergistically and beneficially affect the TGF-β2 induced effects on 2 D and 3D cultured human trabecular meshwork (HTM) cells”. We carefully examined all of the Editor and Reviewer comments and prepared a revised version of our paper that takes these comments into account. The changes are listed below.
Reviewer 3
title: All Trans-Retinoic Acids Synergistically and Beneficially Af- 2 fect the TGF-β2 Induced Effects on 2 D and 3D Cultured Hu- 3 man Trabecular Meshwork (HTM) Cells
It is interesting experimental study on TM cells.
Major comments
- Method is not descriptive, Method 4.2 to be more detail lines 285, Method 4.3 to be more detail. Line 289.
Answer; Thank you for this comment. As suggested, more details of the methods were included “Analyses of the barrier functions of the 2D cultured HTM cells by TEER and FITC-dextran permeability. The barrier functions in the 2D cultured HTM monolayers in the absence and presence of a 5 ng/mL solution of TGF-β2 and/or 10 mM ATRA were evaluated by TEER and FITC dextran permeability measurements as shown in a previous study [30]. Briefly, the TEER values were measured using a 12 mm diameter TEER plate (0.4 μm pore size; Corning Transwell, Sigma-Aldrich) in conjunction with an electrical resistance system (KANTO CHEMICAL CO. INC., Tokyo, Japan). Alternatively, FITC-dextran permeability was estimated by measuring the fluorescence intensity of the amount of FITC that permeated through the membrane over a period of 60 min using a multimode plate reader (Enspire; Perkin Elmer, MA USA) at an excitation wavelength of 490 and an emission wavelength of 530 nm.”, “Analyses of the real time cellular metabolism of the 2D cultured HTM cells by a Seahorse Bioanalyzer. As a real-time cellular metabolic function analysis, OCR and ECAR of the 2D cultured HTM cells in the absence or presence of TGF-β2 (5 ng/mL) and/or ATRA (10 mM) were measured using a Seahorse XFe96 Bioanalyzer (Agilent Technologies, Santa Clara, CA, U.S.A.) as described in a recent report [31-33]. Briefly, 20 x 103 2D HTM cells under several conditions; 1) non-treated control, 2) treated with TGF-β2, 3) treated with ATRA and 4) treated with TGF-β2 and ATRA were grown in wells of a 96-well assay plate. After replacing the culture medium with Seahorse XF DMEM assay medium (pH 7.4, Agilent Technologies, #103575-100) supplemented with 5.5 mM glucose, 2.0 mM glutamine, and 1.0 mM sodium pyruvate, the basal OCR and ECAR values were determined using a Seahorse XFe96 Bioanalyzer and the samples were then further analyzed after supplementation with 2.0 μM oligomycin, 5.0 μM FCCP, 1.0 μM rotenone and antimycin A, and 10mM 2-DG. The OCR and ECAR values were normalized to the amount of protein per well.”, “Configuration, physical properties, size and solidity, analyses of the 3D HTM spheroids. The analyses of the physical properties, mean size and stiffness of the 3D HTM spheroids were performed as described previously [25,34]. Briefly, the mean sizes of the 3D spheroids were directly measured using an inverted microscope (Nikon ECLIPSE TS2; Tokyo, Japan). Alternatively, for the hardness measurements, a single living 3D spheroid was placed on a 3-mm × 3-mm plate and compressed until reaching a deformation of 50 % during a period of 20 seconds using a micro-compressor (MicroSquisher, CellScale, Waterloo, ON, Canada). The force required (μN) was determined, and force/displacement (μN/μm) was calculated.”.
- High Plagiarism rate: issue 19% it should be minimized.
Answer; Thank you for this comment. As suggested, the entire manuscript was revised to avoid a High Plagiarism rate.
- In Discussion: There are numerous conclusions that cannot be drawn from the outcomes of this study. It should contain the analysis of metabolism data and ATRA _TGFB effects, SMA, etc. Association of eye development and vitamin A and immunity of Treg are gap with this study results.
Answer; Thank you for this comment. As suggested, unrelated issues such as the Association of eye development and vitamin A and the immunity of Treg and others were omitted and whole discussion was rewritten in more compact shape;” The action of vitamin A, which is required for the development of the eye, is exerted through an enzymatic metabolite, RA. Within competent cells, RA, which is produced by an autocrine or paracrine mechanism binds to nuclear receptors that are comprised of 2 families of proteins, namely, the RAR α, β, and γ, and the RXR α, β, and γ [41]. The RAR or RXR proteins bind both all-trans- and 9-cis-RA, or only 9-cis-RA which is recognized as the active factor in cell signaling [41].
Among a variety of the biological functions of RA [42], RA has been shown to regulate the development and maintenance of the blood-brain barrier (BBB) [43] as well as the blood- retinal barrier (BRB) [44] by promoting the formation of tight junctions. In terms of the relationship between RA and BRB, Nishikiori et al. reported an interesting result that RAR alpha caused significant reductions in vascular leakage in the diabetic retina by antagonizing the loss of tight junction integrity that is induced by diabetes [45]. In addition to this, RA is also involved in some roles within the anterior chamber of the eye [46] based upon the fact that an analysis using reverse-phase high performance liquid chromatography revealed that the RA level in AH obtained from patients with cataracts was 23.3 +/- 2.3 pmol/ml [47]. Therefore, although these collective findings suggest that RA within AH may influence the biological barrier of the AH drainage route, little has been known related the roles of RA toward TM cells which is the major component of such biological barrier. In the present study, to elucidate these unidentified issues, we initiated several studies as above and found that ATRA exerts synergistic and beneficial effects in TGF-2 treated 2D and 3D cultured HTM cells, which are in vitro models that replicate single and multiple sheet structures of the HTM [25,27,37,40,48]. The findings indicated that the ATRA induced effects observed here were different between the cell culture conditions, 2D and 3D as well as the glaucomatous and none-glaucomatous HTM phenotypes. However, as of this writing, we are unable to confirm the mechanisms responsible for causing such cell culture dependent diversity in terms of the ATRA induced effects. Therefore, to obtain additional information and a better understanding of ATRA-induced effects in glaucomatous and none-glaucomatous HTM, additional studies using RNA sequencing to identify possible factors, and other approaches will be required as our next future project.”
Minor comments:
English editing service required.
Answer; Thank you for this comment. As suggested, English of whole manuscript including some examples as suggested below was edited by native speaking scientific professor Milton S Feather; A letter of confirmation (in PDF format) from him is attached.
For example
- lines 42-43: IOP is well known risk factor for GON, not GON is well known- risk factors
Answer; Thank you for this comment. As pointed out, this was corrected to “IOP is well known risk factor for GON”.
- lines 49-52: “However, and in contrast, such homeostatic regulations are altered in the case of the glaucomatous TM phenotype, in which reduced cellularity and the accumulation of ECM deposits are observed, and the resistance to AH outflow is increased thus resulting in the elevation of IOPs.”
lines 73-75: “However, and in contrast, as of this writing, our knowledge of the drug induced effects of ATRA toward glaucomatous TM, in which TGF-β signaling is also involved in the pathogenesis, remains quite limited.”
The sentence is complex, required scientific expression to be understand.
Answer; Thank you for this comment. Those were corrected to “However, and in contrast, such homeostatic regulations are altered in the case of the glaucomatous TM phenotype. That is, reduced cellularity and the accumulation of ECM deposits within those glaucomatous TM increase the resistance to AH outflow, thus resulting in elevated levels of IOPs [8,9]”, and “However, as of this writing, little is known in terms of the drug induced effects of ATRA on glaucomatous TM, in which TGF-b signaling is also involved in the pathogenesis.”, respectively.
- Abbreviation repetition, line 64 all-trans-retinoic acid (ATRA), line 88 all-trans-retinoic acid (ATRA)
- Lines 77, 91 trans-endothelial electron resistance (TEER)
Answers for 5 and 6; Thank you for this comment. In addition to this issue related to the abbreviations used in the current manuscript, the other reviewer also suggested that we prepare a list of abbreviations rather than to describe abbreviation showed at the first time. Therefore, we prepared such an abbreviation list shown in Table 1.
- Correct expression and need references; Lines 62-73: Based upon these concepts, using TGF-β2 treated TM cells, several in vitro models that replicate the glaucomatous TM phenotype have been reported.
Answer; Thank you for this comment. As suggested, this sentence was changed to “Based upon these observations, several in vitro models using TGF-β2 treated TM cells that replicate the glaucomatous TM phenotype have been reported [16].”, using proper reference; Debra L Fleenor, Allan R Shepard, Peggy E Hellberg, Nasreen Jacobson, Iok-Hou Pang, Abbot F Clark. TGFbeta2-induced changes in human trabecular meshwork: implications for intraocular pressure. Invest Ophthalmol Vis Sci. 2006 Jan;47(1):226-34. doi: 10.1167/iovs.05-1060.
Round 2
Reviewer 2 Report
The authors addressed all of my comments in the response letter and significantly improved the manuscript. However, I suggest the authors consider spelling out the abbreviations when they show up for the first time so that readers can easily understand them. Other than that, I recommend the revised manuscript for publication.
Author Response
Dear Editor,
Thank you very much for the constructive comments concerning our manuscript, " All trans-retinoic acids synergistically and beneficially affect the TGF-β2 induced effects on 2 D and 3D cultured human trabecular meshwork (HTM) cells”. We carefully examined all of the Editor and Reviewer comments and prepared a revised version of our paper that takes these comments into account. The changes are listed below.
Reviewer 2
The authors addressed all of my comments in the response letter and significantly improved the manuscript. However, I suggest the authors consider spelling out the abbreviations when they show up for the first time so that readers can easily understand them. Other than that, I recommend the revised manuscript for publication.
Answer; Thank you for this comment. As suggested, spelling out the abbreviations is included when they show up for the first time, in addition to the abbreviation list.
Reviewer 3 Report
title: All Trans-Retinoic Acids Synergistically and Beneficially Af- 2 fect the TGF-β2 Induced Effects on 2 D and 3D Cultured Hu- 3 man Trabecular Meshwork (HTM) Cells
High Plagiarism rate: issue was improved after major revision & English is improved.
However, I think it is an issue since it is not able to view the contents of the discussion, such as the comparison of the results of prior studies, etc. connected to the relevance, limitations, and research outcomes of the study.
Author Response
Dear Editor,
Thank you very much for the constructive comments concerning our manuscript, " All trans-retinoic acids synergistically and beneficially affect the TGF-β2 induced effects on 2 D and 3D cultured human trabecular meshwork (HTM) cells”. We carefully examined all of the Editor and Reviewer comments and prepared a revised version of our paper that takes these comments into account. The changes are listed below.
Reviewer 3
title: All Trans-Retinoic Acids Synergistically and Beneficially Affect the TGF-β2 Induced Effects on 2 D and 3D Cultured Human Trabecular Meshwork (HTM) Cells
High Plagiarism rate: issue was improved after major revision & English is improved.
However, I think it is an issue since it is not able to view the contents of the discussion, such as the comparison of the results of prior studies, etc. connected to the relevance, limitations, and research outcomes of the study.
Answer; Thank you for this comment. As suggested, Discussion was rewritten to include the contents of the discussion, such as the comparison of the results of prior studies, etc. connected to the relevance, limitations, and research outcomes of the study as much as possible; “The action of vitamin A, which is required for the development of the eye, is exerted through an enzymatically produced metabolite, RA (retinoic acid). Within competent cells, RA, which is produced by an autocrine or paracrine mechanism binds to nuclear receptors that are comprised of 2 families of proteins, namely, the retinoic acid receptors (RAR) α, β, and γ, and the retinoid X receptors (RXR) α, β, and γ [25]. The RAR or RXR proteins both bind all-trans- and 9-cis-RA, or only 9-cis-RA which is recognized as the active factor in cell signaling [25]. As pathophysiological roles of RA, RA suppresses TGF-β–induced fibrogenic changes in various tissues including the conjunctiva [20, 22, 23]. In addition to the well-known fact that TGF-β2 is involved in the pathogenesis of glaucomatous TM [16], bioinformatic prioritization and functional annotation of genome-wide association studies (GWASs)-based candidate genes for POAG identified that RA receptor signaling in addition to TGF-β, blood vessel development were the most significant pathways related to ECM metabolism [26] It was also reported that RA alone or when combined with a glucocorticoid induced a significant up-regulation in the expression of myocilin within immortalized TM cells, suggesting that RA signaling may also be involved in the pathogenesis of steroid-induced glaucoma (SG) related TM [27]. These collective observations strongly suggest that RA signaling may have significant roles within the pathogenesis of glaucomatous TM. In fact, RA is identified within the AH [28] based upon the fact that an analysis using reverse-phase high performance liquid chromatography revealed that the RA level in AH obtained from patients with cataracts was 23.3 +/- 2.3 pmol/ml [29]. In the current study, we report that ATRA on glaucomatous human TM by using 2D and 3D HTM cell culture models based upon following results; 1) ATRA caused the inhibition of the TGF-β2-induced increase in barrier functions and metabolic reserve from mitochondrial oxidative phosphorylation to glycolysis in 2D HTM cells, and 2) ATRA caused a substantial downsizing and softenening of 3D HTM spheroids in the absence and presence of TGF-β2. We therefore conclude that this is the first demonstration of the existence of ATRA induced effects 2D and 3D cultured HTM cells that had been treated with on TGF-β2, as the in vitro glaucomatous TM models, although those effects varied between the 2D and 3D cultures.
As of this writing, we are unable to report on the details of the underlying mechanisms that cause such RA induced effects toward the glaucomatous TM, especially their function as a biological barrier to AH outflow. However, among the many biological functions of RA [30], RA has been shown to regulate the development and maintenance of the blood-brain barrier (BBB) [31] as well as the blood- retinal barrier (BRB) [32] by promoting the formation of tight junctions. In addition, Nishikiori et al. reported an interesting result concerning the relationship between RA and BRB. That is, RAR alpha caused significant reductions in vascular leakage in diabetic retinas by antagonizing the loss of tight junction integrity that is induced by diabetes [33]. These observations are consistent with our current results showing that ATRA exerts synergistic and beneficial effects in TGF-β2 treated 2D and 3D cultured HTM cells, which are in vitro models that replicate single and multiple sheet barrier structures of the HTM [34-38].
However, the current study also has the following limitations; we do not know 1) why the ATRA induced effects observed here were different between the cell culture conditions, and 2) which signaling is affected by ATRA within the non-glaucomatous and glaucomatous TM. Interestingly, recent observations that the activation of the RA signaling pathway could suppress pseudoexfoliation (PEX)-associated ECM production and the formation of microfibrillar networks via antagonizing Smad-dependent TGF-β1 signaling [39] may provide significant clues to answer these questions. Therefore, to obtain additional information and a better understanding of ATRA-induced effects in none-glaucomatous and glaucomatous HTM including POAG, SG and PEX, additional studies using RNA sequencing to identify possible factors, and other approaches will be required as our next future project.
In conclusion, our current, novel findings demonstrate that ATRA may induce synergistic and beneficial effects on TGF-β2 treated 2D and 3D cultured HTM cells, providing fundamental ideas for possible therapeutic approaches by modulating RA signaling within glaucoma.”
Round 3
Reviewer 3 Report
After major revision, the paper may be acceptable to reviewer's consideration.
Author Response
Dear Editor,
Thank you very much for the constructive comments concerning our manuscript, " All trans-retinoic acids synergistically and beneficially affect the TGF-β2 induced effects on 2 D and 3D cultured human trabecular meshwork (HTM) cells”. We carefully examined all of the Editor and Reviewer comments and prepared a revised version of our paper that takes these comments into account. The changes are listed below.
Academic editor comment
This is an interesting study. Retinoic acid is a metabolite of vitamin A1 and the major mediators of RA signaling are the retinoic acid receptor (RAR) and the retinoid X receptor (RXR), which are ligand-dependent transcription factors belonging to the nuclear receptor superfamily. Please introduce more nuclear receptors related background in the introduction or discussion to match the theme this special issue.
Answer; Thank you for this comment. As suggested, we emphasize “nuclear receptor” within 1st sentence of the 2nd paragraph of introduction; “A derivative of vitamin A, all-trans-retinoic acid (ATRA) is a potent regulator of the growth and differentiation of various types of cells via RA nuclear receptors [17,18].” and discussion, 1st paragraph “RA and its natural and synthetic derivatives, retinoids, are pivotal dietary factors regulating cellular differentiation and growth by binding nuclear receptors that are comprised of 2 families of proteins, namely, the retinoic acid receptors (RAR) α, β, and γ, and the retinoid X receptors (RXR) α, β, and γ [41] (ref; Curr Med Chem. 2006;13(29):3553-63. doi: 10.2174/092986706779026183.). These ligand-activated nuclear receptors stimulate the transcription of target genes by binding to RA-responsive elements within the promoter regions. The RAR or RXR proteins bind both all-trans- and 9-cis-RA, or only 9-cis-RA which is recognized as the active factor in cell signaling [41]. Such RA induced nuclear receptor regulations are also required for physiological and pathological events of ocular tissues. As pathophysiological roles of RA, RA suppress the TGF-β–induced fibrogenic changes of various tissues including conjunctiva [20][22,23]. In addition to the well known fact that TGF-b2 is involved in the pathogenesis of the glaucomatous TM [16], bioinformatic prioritization and functional annotation of genome-wide association studies (GWASs)-based candidate genes for POAG identified that RA nuclear receptor signaling in addition to TGF-β, blood vessel development were the most significant pathways related to ECM metabolism (ref; Genes (Basel). 2022 Jun 13;13(6):1055. doi: 10.3390/genes13061055). In addition, it was reported that RA alone or combined with a glucocorticoid induced significant up-regulation of myocilin expression within immortalized TM cells, suggesting that RA signaling may also be involved in the pathogenesis of the steroid-induced glaucoma (SG) related TM (Exp Eye Res. 2017 Feb;155:91-98. doi: 10.1016/j.exer.2017.01.006. Epub 2017 Jan 30.). These collective observations strongly suggested that RA signaling may have significant roles within the pathogenesis of the glaucomatous TM. In fact, RA is identified within the AH [46] based upon the fact that an analysis using reverse-phase high performance liquid chromatography revealed that the RA level in AH obtained from patients with cataracts was 23.3 +/- 2.3 pmol/ml [47]. In the current study, we demonstrated the significant effects of ATRA toward the glaucomatous human TM by using 2D and 3D HTM cell culture models based upon following results; 1) ATRA caused inhibition of the TGF-β2-induced increase of the barrier functions and metabolic reserve from mitochondrial oxidative phosphorylation to glycolysis of the 2D HTM cells, and 2) ATRA substantially downsized and softener 3D HTM spheroids in the absence and presence of TGF-b2. Therefore, we believed that this is the first demonstration of the ATRA induced effects on TGF-b2 treated 2D and 3D cultured HTM cells, as the in vitro glaucomatous TM models, although those effects varied between the 2D and 3D cultures.”, and last paragraph “In conclusion, our current findings demonstrated a novel finding that ATRA may induce synergistic and beneficial effects on TGF-b2 treated 2D and 3D cultured HTM cells, providing fundamental ideas to seek a possibility of therapeutic prospect by modulating RA related nuclear receptor actions within glaucoma.”.
